# α-Trifluoromethyl Chalcones as Potent Anticancer Agents for Androgen Receptor-Independent Prostate Cancer

**DOI:** 10.3390/molecules26092812

**Published:** 2021-05-10

**Authors:** Yohei Saito, Atsushi Mizokami, Kouji Izumi, Renato Naito, Masuo Goto, Kyoko Nakagawa-Goto

**Affiliations:** 1School of Pharmaceutical Sciences, College of Medical, Pharmaceutical and Health Science, Kanazawa University, Kanazawa 920-1192, Japan; saito-y@staff.kanazawa-u.ac.jp; 2Department of Integrative Cancer Therapy and Urology, School of Medical Sciences, Kanazawa University, Kanazawa 920-1192, Japan; kouji1974@staff.kanazawa-u.ac.jp (K.I.); thealfuu@yahoo.co.jp (R.N.); 3Chemical Biology and Medicinal Chemistry, Eshelman School of Pharmacy, University of North Carolina, Chapel Hill, NC 27599, USA; goto@med.unc.edu

**Keywords:** chalcone, trifluoromethyl group, prostate cancer, antitumor activity, target proteins

## Abstract

α-Trifluoromethyl chalcones were prepared and evaluated for their antiproliferative activities against androgen-independent prostate cancer cell lines as well as five additional types of human tumor cell lines. The most potent chalcone **5** showed superior antitumor activity in vivo with both oral and intraperitoneal administration at 3 mg/kg. Cell-based mechanism of action studies demonstrated that **5** induced cell accumulation at sub-G1 and G2/M phases without interfering with microtubule polymerization. Furthermore, several cancer cell growth-related proteins were identified by using chalcone **5** as a bait for the affinity purification of binding proteins.

## 1. Introduction

A small change in a chemical structure sometimes induces a big change in a biological profile. The features of substituents on a bioactive molecule influence the various interactions between it and its target protein and subsequently affect its selectivity and potency, as we have previously reported [1,2]. Fluorine’s unique properties, such as small atomic size and high electronegativity, impact pKa, molecular conformation, binding affinity with target molecules, membrane permeability, metabolic pathways, and pharmacokinetic properties of bioactive molecules. Accordingly, fluorine attracts attention in drug discovery and development [3]. In our continuing structure–activity relationship (SAR) study of chalcone derivatives as anti-prostate cancer agents [4,5], we explored whether the insertion of fluorine and/or a fluorinated functional group into synthetic chalcones would improve their activity. Chalcone, a biosynthetic intermediate of various flavonoids, is composed of two aromatic rings connected by α,β-unsaturated carbonyl unit. Initially, we introduced a trifluoromethyl (CF_3_) group, a known electron withdrawing group, at the α-position of the olefin. This group might accelerate Michael addition of a nucleophile in biomolecules, such as the SH in cysteine [6]. CF_3_ has been also used as an isostere of a methyl group or amide C=O bond with the aim of improving metabolic stability and hydrophobicity in drug development [7]. Secondly, we introduced electron withdrawing groups on the aromatic ring connected to the olefinic double bond of the α-CF_3_ chalcones. The synthesized α-CF_3_ analogues were evaluated for anti-prostate cancer activity.

In many cases, prostate cancer progresses slowly and is generally controllable by hormone therapy and/or castration. However, the cancer can reoccur after several years and then is termed castration-resistant prostate cancer (CRPC). Currently, no effective treatment exists for CRPC. Prostate cancer is associated with androgen-receptors (ARs), and prostate cancer cells grow in response to androgen. While hormone therapy suppresses androgen secretion and restrains cancer cell growth, continued treatment for several years often leads to diminished effectiveness due to the induction of mutations on the AR, which then can be activated by only a small amount of androgen or independently of constitutive active ligand [8]. In addition, the emergence of prostate cancer that lacks AR expression accelerates its malignancy. In this study, the synthesized α-CF_3_ chalcones were assayed against DU145 and PC-3 cell lines, which do not express ARs, to evaluate their effectiveness against CRPC. The most potent derivative was further investigated in mode of action studies, including identification of target proteins, and for in vivo antitumor effects. The differences of intracellular targets of previously reported α-substituted chalcones [9,10,11] were also discussed.

## 2. Results and Discussion 

### 2.1. Chemistry

The novel α-trifluoromethyl chalcones (**2**–**7**) and known **1 [12]**, were synthesized from the related chalcones [13,14,15,16] including novel **10** and **11**, which were obtained by Claisen-Schmidt condensation of aryl methyl ketones and aromatic aldehydes. In target compounds **1**–**7**, the aromatic aldehyde was either unsubstituted or substituted with one of three electron withdrawing groups (F, CF_3_, or NO_2_) or an electron donating group (NMe_2_) for comparative purposes. The CF_3_ moiety was inserted at the α-position of the chalcones by using 1-trifluoromethyl-1,2-benziodoxol-3(1*H*)-one (Togni reagent) [17] in the presence of CuI at 80 °C using a reported method (Scheme 1) [18]. In **6**, the chalcone ring-A was changed from benzene to naphthalene and, in derivative **7**, the chalcone ring-B was replaced with benzothiophene, since potent antiproliferative effects were observed with other flavonoids containing 10π-electron aromatic ring systems [1,19,20]. It should be noted that the conformation of the olefin in the resulting α-CF_3_ chalcones is *cis* rather than *trans* as in the natural chalcone, which has been confirmed with X-ray crystal structure analysis by Bi et al. [18] and might also affect the biological activities. Spectroscopic data of all newly synthesized compounds are in Appendix A.

### 2.2. Biological Evaluation

#### Antiproliferative Activity of Compounds against AR-Independent Cells

The antiproliferative activities of the synthesized α-CF_3_ chalcones against AR-independent cell lines, DU145 and PC-3 were evaluated (Table 1). 5′-Chloro-2,2′-dihydroxychalcone (Cl-DHC), which was found as a potent antiproliferative chalcone by our group [5], was used as a control. All α-CF_3_ chalcones showed potent activity; especially 4-NO_2_ chalcone **2** and 3,4-difluorochalcone **5** strongly inhibited the growth of both tumor cell lines with IC_50_ values of less than 0.2 μM. These results indicated the insertion of CF_3_ at the α-position was beneficial to the antiproliferative activity, since most of the potent compounds among our previously synthesized chalcones without an α-CF_3_ [4,5] exhibited IC_50_ values of over 5 μM. Regarding the chalcone ring-A, although the α-CF_3_ chalcone **6** with a naphthyl ring-A was active, it was threefold less potent than the analogous chalcone **5** with a phenyl ring-A. Among this limited compound set, electron withdrawing groups on ring-B resulted in slightly improved antiproliferative activity, as the non-substituted chalcone **1** and 4-NMe_2_ chalcone **3** were less potent than 4-NO_2_ **2**, 4-CF_3_ **4**, and 3,4-difluoro **5**.

All synthesized chalcones were also assayed against five additional human tumor cell lines, non-small cell lung (A549), triple-negative breast (MDA-MB-231), estrogen-responsible breast (MCF-7), cervical cancer cell line HeLa derivative (KB), and P-glycoprotein (P-gp) overexpressing multidrug resistant KB subline (KB-VIN). Again, the chalcones containing electron withdrawing group on ring-B (**2**, **4**, and **5**) showed more potent antiproliferative activity than the remaining compounds (Table 2). In addition, KB and KB-VIN cells displayed similar susceptibility to the tested chalcones. This finding suggested that the compounds might target specific proteins that are critical for cell growth and overexpressed in KB and KB-VIN cells.

The resistance of tumor cells to drugs is always a severe obstacle to effective chemotherapy. As shown in Table 2, all tested compounds showed similar antiproliferative activity against KB and KB-VIN, suggesting that our compounds were not affected by the drug transporter P-gp. The most promising chalcone **5** was further tested against four kinds of taxane-resistant prostate cancer cell lines, DU145/TxR (docetaxel resistant DU145), DU145/TxR/CxR (docetaxel and cabazitaxel resistant DU145), PC-3/TxR (docetaxel resistant PC-3), and PC-3/TxR/CxR (docetaxel and cabazitaxel resistant PC-3) [21]. Chalcone **5** showed significant antiproliferative activity against these cells with IC_50_ values of 0.14–0.28 μM (Table 3). Taken together, the results suggesting that chalcone **5** potentially overcomes castration and taxane resistances. 

To estimate the in vivo antitumor effects of chalcone **5**, we tested it in a xenograft antitumor model assay using PC-3. As anticipated, the tumor growth was efficiently suppressed with both intraperitoneal and oral administration of **5** without significant weight loss compared with control (Figure 1). Notably, a dose of only 3 mg/kg was used in this study, even though many reported studies have used much larger doses of test compounds. 

The identification of target proteins is an important step in drug discovery and development. Accordingly, to study the mode of action of our new chalcones, we performed affinity labeling using a specifically designed and synthesized chemical probe (**9**) from the novel α-CF_3_ chalcone **8**. 4′-Amino-3,4-difluorochalcone [22] was selected for the introduction of propargyl group, which would work as a tether to immobilize on azide beads. Trifluoromethylation of 4′-amino-3,4-difluorochalcone gave **8** and propargylation of the amino group on **8** provided the target chalcone **9** (Scheme 2). The antiproliferative activities of **9** against DU145 and PC-3 were 1.43 and 1.34 μM, respectively (Table 4). Thus, chalcone **9** was 1.9–2.3 fold more potent than **8** (IC_50_ 3.25 μM for DU145 and 2.48 μM for PC-3). In flow cytometric analysis, both **8** and **9** showed insignificant effects against KB-VIN despite using a three-fold concentration of their IC_50_ value. However, the treated MDA-MB-231 cells clearly showed accumulations of sub-G1 phase, a typical apoptotic pattern, and G2/M phase cells (Figure 2a).

Compared with the parent molecule **5**, chalcones **8** and **9** showed diminished antiproliferative effects against all tested human tumor cell lines (Table 3 and Table 4). However, all three compounds displayed the same distribution pattern using flow cytometric analysis (Figure 2a), indicating the same intracellular targets. The tubulin polymerization inhibitor combretastatin A-4 (CA-4) was also evaluated in this study, since antiproliferative chalcones often target tubulin [23,24]. However, the effects of chalcone **5** on the cell cycle progression were clearly distinguishable from those of CA-4 (Figure 2a). Because G2/M phase accumulation often results from treatment with tubulin inhibitors [25], we evaluated the effects of chalcone **5** on tubulin polymerization and cell morphology by using immunocytochemical studies with an antibody to α-tubulin (Figure 2b). The resulting confocal images indicated that chalcone **5** did not significantly affect either tubulin polymerization or cell morphology. The above cell-based mode of action studies suggested that chalcone **5** targets proteins related to the cell cycle progression in G2/M and apoptotic induction. Therefore, we next conducted affinity purification of chalcone **5**-binding proteins using beads conjugated to compound **9** as a bait to identify the target proteins.

Since **5** and **9** likely have identical intracellular mechanisms of action, affinity purification of the intracellular binding proteins of **5** was carried out using compound **9**, the propargylated probe of **5**. Compound **9** was immobilized on azide beads via a click reaction, and the obtained beads were reacted with PC-3 cell lysate followed by elution of binding proteins using SDS-PAGE sample buffer. The eluted fractions were separated by SDS-polyacrylamide gel electrophoresis (SDS-PAGE). Five main bands were detected by staining with Coumassie Brilliant Blue (CBB) dye, corresponding to proteins with molecular weights between 100 and 40 kDa (Figure 3). Since all bands became undetectable when the lysate was preincubated with compound **5** as a competitive inhibitor, the bands were cut out and the proteins were digested by adding trypsin to the gel. The obtained trypsin-digested peptide sequences were identified by LC-MS/MS analysis. HSP90 and MICOS complex subunit MIC60 were detected from band No. 1. HSP70 (band No. 2), HSP60 (band No. 3), pyruvate kinase (band No. 3), T-complex protein 1 subunits (band No. 3), alpha-enolase (band No. 4), actin (band No. 5), and proliferation-associated protein 2G4 (band No. 5) were also identified. The list of identified proteins is shown in Appendix A). Heat shock proteins (HSPs) are well known not only as significant factors for cancer development but also targets for cancer therapy [26]. Glycolytic enzymes, including pyruvate kinase and alpha-enolase, are also related to tumor growth [27] and reported as diagnostic targets [28]. The unique target MIC60 plays an important role in the maintenance of mitochondrial crista structure [29] and PA2G4 isoforms display opposing functions, i.e., suppressive and promoting effects, in cancer [30]. Compound **5** might suppress cancer cell proliferation mainly by interacting with these proteins. Further studies are merited to demonstrate the clear target of chalcone **5** followed by the elucidation of druggable targets for drug development against CRPC. We are currently performing gene knockdown studies of the expressions of the identified proteins using siRNA in PC-3 cells.

## 3. Materials and Methods

### 3.1. Chemistry

All solvents and chemicals were used as purchased without further purification. The progress of reactions was monitored on Merck Millipore precoated silica gel glass plates (TLC Silica gel 60 F254). Column chromatography was performed with Kanto chemical silica gel 60 N (spherical, neutral) or preparative TLC was carried out with Merck Millipore precoated silica gel glass plates (PLC Silica gel 60 F254, 1 mm) for the purification of synthetic compounds. ^1^H and ^13^C NMR spectra were recorded on JEOL JNM-ECA 600 or JNM-ECS 400 using CDCl_3_ as solvent and referenced to TMS or residual solvent peak. Chemical shifts δ are reported in ppm. Mass spectrometric analysis were performed using JEOL JMS-700 Mstation or JMS-T100TD. UV spectra were recorded on a Tecan SPARK 10 M in MeCN-H_2_O (1:1). IR spectra were measured on Shimadzu IRSprit in neat. Melting points were measured on an AS ONE ATM-02. The purity of all compounds was determined as >95% by ^1^H NMR. 

### 3.2. General Procedures for Chalcones

To a solution of 1′-acetonaphthone (120 mg, 0.70 mmol) and 3,4-difluorobenzaldehyde (100 mg, 0.70 mmol) in EtOH (1.0 mL) was added 40% aqueous KOH (0.5 mL), and the mixture was stirred at room temperature. After the reaction was complete, ice-water was added to the reaction mixture, which was then neutralized with 1N HCl. The mixture was extracted with EtOAc and the resultant organic phase was washed with brine, dried over MgSO_4_, and concentrated in vacuo. The residue was purified by column chromatography on SiO_2_, eluting with hexane-EtOAc (15:1) to give (*E*)-3-(3,4-difluorophenyl)-1-(naphthalen-1-yl)prop-2-en-1-one (172 mg, 0.58 mmol) in 83% yield as a colorless solid. All chalcones were produced by the same procedure.

#### 3.2.1. (*E*)-3-(3,4-difluorophenyl)-1-(naphthalen-1-yl)prop-2-en-1-one (**10**)

Rf 0.25 (hexane/EtOAc, 10:1); mp 81 °C; UV (MeCN/H_2_O, 1:1) λ_max_ (nm) 220, 302; IR (neat) ν_max_ (cm^−1^) 3050, 1652, 1583, 1516, 1271, 1097, 980, 796, 772; ^1^H NMR (600 MHz, CDCl_3_): δ 8.33 (d, *J* = 7.8 Hz, 1H), 8.01 (d, *J* = 8.4 Hz, 1H), 7.92 (d, *J* = 6.6 Hz, 1H), 7.77 (d, *J* = 6.6 Hz, 1H), 7.59–7.51 (m, 4H), 7.40 (ddd, *J* = 10.2, 8.4, 1.8 Hz, 1H), 7.30 (ddd, *J* = 6.6, 4.2, 1.8 Hz, 1H), 7.22 (d, *J* = 16.2 Hz, 1H), 7.19 (m 1H); ^13^C NMR (150 MHz, CDCl_3_): δ 194.9, 151.7 (dd, *J* = 252.8, 12.9 Hz), 150.7 (dd, *J* = 248.6, 12.9 Hz), 143.2, 136.7, 133.9, 132.0, 131.9 (dd, *J* = 4.4, 4.4 Hz), 130.4, 128.5, 127.8, 127.6, 127.3, 126.6, 125.5, 125.3 (dd, *J* = 5.7, 2.9 Hz), 124.5, 117.9 (d, *J* = 18.6 Hz), 116.6 (d, *J* = 18.6 Hz); HRMS (FAB) *m/z*: [M+H]^+^ calcd for C_19_H_13_F_2_O 295.0934, found 295.0941.

#### 3.2.2. (*E*)-3-(benzo[*b*]thiophen-3-yl)-1-phenylprop-2-en-1-one (**11**)

The chalcone was prepared from acetophenone and benzo[b]thiophene-3-carboxaldehyde as a yellow solid. Yield: 70%. Rf 0.3 (hexane/EtOAc, 10:1); mp 94–95 °C; UV (MeCN/H_2_O, 1:1) λ_max_ (nm) 226, 266, 356; IR (neat) ν_max_ (cm^−1^) 3091, 3055, 1653, 1586, 1575, 1495, 1367, 1202, 1010, 971, 780, 729, 690; ^1^H NMR (400 MHz, CDCl_3_): δ 8.15–8.05 (m 4H), 7.93 (s, 1H), 7.92 (d, *J* = 7.2 Hz, 1H), 7.66 (d, *J* = 16.0 Hz, 1H), 7.61 (m, 1H), 7.56–7.49 (m, 3H), 7.44 (dd, *J* =7.2, 7.2 Hz, 1H); ^13^C NMR (150 MHz, CDCl_3_): δ 190.4, 140.6, 138.2, 137.3, 136.4, 132.8, 132.3, 128.7, 128.7, 128.5, 125.2, 125.1, 123.1, 122.4, 122.2; HRMS (FAB) *m/z*: [M+H]^+^ calcd for C_17_H_13_OS 265.0687, found 265.0684.

### 3.3. General Procedures for a-CF_3_ Chalcones 

To a mixture of 1-trifluoromethyl-1,2-benziodoxol-3(1*H*)-one (170 mg, 0.22 mmol) and copper iodide (2.7 mg, 0.014 mmol) was added chalcone (30 mg, 0.14 mmol) in DMF (0.2 mL) under N_2_ atmosphere and the mixture was stirred at 80 °C. After the reaction was complete, the mixture was concentrated. The residue was purified by column chromatography on silica gel, eluting with hexane-EtOAc (50:1) to give (*E*)-1,3-diphenyl-2-(trifluoromethyl)prop-2-en-1-one (**1**) [9] (9.4 mg, 0.034 mmol) in 24% yield as a pale, yellow oil. α-CF_3_ chalcones **2**–**8** were produced by the same procedure.

#### 3.3.1. (*E*)-3-(4-nitrophenyl)-1-phenyl-2-(trifluoromethyl)prop-2-en-1-one (**2**) 

Compound **2** was prepared from 4-nitrochalcone [10] as a colorless solid. Yield: 9%. Rf 0.25 (hexane/EtOAc, 10:1); mp 119–120 °C; UV (MeCN/H_2_O, 1:1) λ_max_ (nm) 262; IR (neat) ν_max_ (cm^−1^) 3079, 2933, 2859, 1667, 1596, 1520, 1346, 1267, 917, 824, 687; ^1^H NMR (600 MHz, CDCl_3_): δ 8.06 (d, *J* = 8.4 Hz, 2H), 7.87 (dd, *J* = 8.4, 1.2 Hz, 2H), 7.56 (tt, *J* = 7.2, 1.2 Hz, 1H), 7.54 (s, 1H), 7.43 (d, *J* = 9.0 Hz, 2H), 7.40 (t, *J* = 7.8 Hz, 2H); ^13^C NMR (125 MHz, CDCl_3_): δ 192.7, 148.3, 138.2, 135.0, 134.9, 134.1 (q, *J* = 5.9 Hz), 130.1, 129.6, 129.1, 123.9; HRMS (FAB) *m/z*: [M+H]^+^ calcd for C_16_H_11_F_3_NO_3_ 322.0691, found 322.0678.

#### 3.3.2. (*E*)-3-[4-(dimethylamino)phenyl]-1-phenyl-2-(trifluoromethyl)prop-2-en-1-one (**3**)

Compound **3** was prepared from 4-(dimethylamino)chalcone [11] as a yellow oil. Yield: 26%. Rf 0.2 (hexane/EtOAc, 10:1); UV (MeCN/H_2_O, 1:1) λ_max_ (nm) 254, 336, 398; IR (neat) ν_max_ (cm^−1^) 3060, 2909, 2861, 2816, 2359, 1660, 1593, 1526, 1278, 1147, 1108, 814, 696, 687; ^1^H NMR (400 MHz, CDCl_3_): δ 7.96 (dd, *J* = 8.4, 1.2 Hz, 2H), 7.52 (tt, *J* = 7.2, 1.2 Hz, 1H), 7.39 (t, *J* = 8.4 Hz, 2H), 7.34 (d, *J* = 1.2 Hz, 1H), 7.11 (d, *J* = 9.2 Hz, 2H), 6.44 (d, *J* = 9.2 Hz, 2H), 2.91 (s, 6H); ^13^C NMR (100 MHz, CDCl_3_): δ 193.9, 151.3, 137.3 (q, *J* = 5.7 Hz), 136.0, 134.0, 131.8, 129.7, 128.8, 119.3, 111.4, 39.9; HRMS (FAB) *m/z*: [M+H]^+^ calcd for C_18_H_17_F_3_NO 320.1262, found 320.1262.

#### 3.3.3. (*E*)-1-phenyl-2-(trifluoromethyl)-3-(4-(trifluoromethyl)phenyl)prop-2-en-1-one (**4**)

Compound **4** was prepared from 4-(trifluoromethyl)chalcone [12] as a pale-yellow oil. Yield: 9%. Rf 0.4 (hexane/EtOAc, 10:1); UV (MeCN/H_2_O, 1:1) λ_max_ (nm) 254; IR (neat) ν_max_ (cm^−1^) 3067, 2928, 2853, 1670, 1323, 1269, 1163, 1067, 834, 687; ^1^H NMR (400 MHz, CDCl_3_): δ 7.89 (dd, *J* = 8.4, 1.2 Hz, 2H), 7.56 (tt, *J* = 7.2, 1.2 Hz,1H), 7.51 (s, 1H), 7.47 (d, *J* = 8.4 Hz, 1H, H-Ar), 7.42–7.36 (m, 4H); ^13^C NMR (150 MHz, CDCl_3_): δ 192.0, 135.4, 135.1, 135.0 (q, *J* = 5.7 Hz), 134.7, 131.8 (q, *J* = 33.0 Hz), 131.4 (q, *J* = 30.2 Hz), 129.6, 129.6, 129.0, 125.8 (q, *J* = 4.2 Hz), 123.5 (q, *J* = 270.0 Hz), 121.9 (q, *J* = 274.4 Hz); HRMS (FAB) *m/z*: [M+H]^+^ calcd for C_17_H_11_F_6_O 345.0714, found 345.0716.

#### 3.3.4. (*E*)-3-(3,4-difluorophenyl)-1-phenyl-2-(trifluoromethyl)prop-2-en-1-one (**5**) 

Compound **5** was prepared from 3,4-diifluorochalcone [13] as a colorless oil. Yield: 29%. Rf 0.3 (hexane/EtOAc, 10:1); UV (MeCN/H_2_O, 1:1) λ_max_ (nm) 254; IR (neat) ν_max_ (cm^−1^) 3064, 2918, 1667, 1517, 1263, 1119, 917, 686; ^1^H NMR (600 MHz, CDCl_3_): δ 7.89 (dd, *J* = 8.4, 1.2 Hz, 2H), 7.57 (t, *J* = 7.2 Hz, 1H), 8.16 (t, *J* = 7.8 Hz, 2H), 7.39 (s, 1H), 7.10–7.07 (m, 1H), 7.03–6.98 (m, 2H); ^13^C NMR (150 MHz, CDCl_3_): δ 192.1, 151.2 (dd, *J* =252.8, 12.9 Hz), 150.1 (dd, *J* = 249.9, 12.9 Hz), 135.1, 134.8, 134.4 (q, *J* = 5.7 Hz), 130.1 (q, *J* = 30.2 Hz), 129.8, 129.6, 129.0, 126.2, 122.0 (q, *J* = 273.0 Hz), 118.4 (d, *J* = 18.6 Hz), 117.9 (d, *J* = 18.8 Hz); HRMS (FAB) *m/z*: [M+H]^+^ calcd for C_16_H_10_F_5_O 313.0652, found 313.0653.

#### 3.3.5. (*E*)-3-(3,4-difluorophenyl)-1-(naphthalen-1-yl)-2-(trifluoromethyl)prop-2-en-1-one (**6**)

Compound **6** was prepared from (*E*)-3-(3,4-difluorophenyl)-1-(naphthalen-1-yl)prop-2-en-1-one **10** as a yellow solid. Yield: 16%. Rf 0.35 (hexane/EtOAc, 10:1); mp 71–72 °C; UV (MeCN/H_2_O, 1:1) λ_max_ (nm) 212, 332; IR (neat) ν_max_ (cm^−1^) 3087, 3063, 2919, 1655, 1516, 1272, 1243, 1229, 1122, 1110, 1001, 920, 776; ^1^H NMR (600 MHz, CDCl_3_): δ 9.05 (d, *J* = 9.0 Hz, 1H), 8.01 (d, *J* = 8.4 Hz, 1H), 7.92 (d, *J* = 7.2 Hz, 1H), 7.88 (d, *J* = 7,8 Hz, 1H), 7.73 (t, *J* = 7.8 Hz, 1H), 7.60 (t, *J* = 7.2 Hz, 1H), 7.43 (s, 1H), 7.38 (t, *J* = 7.8 Hz, 1H), 7.12–7.09 (m, 1H), 7.04–7.01 (m, 1H), 6.87–6.83 (m, 1H); ^13^C NMR (150 MHz, CDCl_3_): δ 193.3, 151.0 (dd, *J* = 252.8, 12.9 Hz), 150.1 (dd, *J* = 248.6, 12.9 Hz), 135.6, 135.2 (q, *J* = 5.7 Hz), 133.9, 132.7, 132.2 (q, *J* = 30.2 Hz), 132.0, 130.9, 129.4, 129.3, 128.7, 127.0, 126.0, 125.4, 124.2, 122.2 (q, *J* = 272.9 Hz), 118.3 (d, *J* = 18.6 Hz), 117.6 (d, *J* = 17.3 Hz); HRMS (FAB) *m/z*: [M+H]^+^ calcd for C_20_H_12_F_5_O 363.0808, found 363.0793.

#### 3.3.6. (*E*)-3-(benzo[*b*]thiophen-3-yl)-1-phenyl-2-(trifluoromethyl)prop-2-en-1-one (**7**)

Compound **7** was prepared from (*E*)-3-(benzo[b]thiophen-3-yl)-1-phenylprop-2-en-1-one **11** as a colorless oil. Yield: 20%. Rf 0.4 (hexane/EtOAc, 10:1); UV (MeCN/H_2_O, 1:1) λ_max_ (nm) 228, 266, 316; IR (neat) ν_max_ (cm^−1^) 3064, 3031, 1663, 1335, 1272, 1169, 1120, 910, 757, 732, 686, 589; ^1^H NMR (600 MHz, CDCl_3_): δ 7.90–7.87 (m, 3H), 7.78 (d, *J* = 7.8 Hz, 1H), 7.74 (s, 1H), 7.48 (dt, *J* = 7.2, 1.2 Hz, 2H), 7.40 (t, *J* = 7.8 Hz, 1H), 7.37 (s, 1H), 7.33 (t, *J* = 7.2 Hz, 2H); ^13^C NMR (150 MHz, CDCl_3_): δ 192.8, 139.5, 137.6, 135.2, 134.4, 129.6, 129.4, 128.8, 128.1 (q, *J* = 5.7 Hz), 127.6, 125.2, 125.0, 122.9, 121.2; HRMS (FAB) *m/z*: [M+H]^+^ calcd for C_18_H_12_F_3_OS 333.0561, found 333.0574.

#### 3.3.7. (*E*)-1-(4-aminophenyl)-3-(3,4-difluorophenyl)-2-(trifluoromethyl)prop-2-en-1-one (**8**) 

Compound **8** was prepared from (*E*)-1-(4-aminophenyl)-3-(3,4-difluorophenyl)prop-2-en-1-one [19] as a yellow solid. Yield: 53%. Rf 0.2 (hexane/EtOAc, 3:1); mp 94–95 °C; UV (MeCN/H_2_O, 1:1) λ_max_ (nm) 248, 350; IR (neat) ν_max_ (cm^−1^) 3422, 3345, 3232, 3963, 2919, 1636, 1580, 1558, 1523, 1300, 1256, 1173, 1116; ^1^H NMR (400 MHz, CDCl_3_): δ 7.73 (d, *J* = 8.8 Hz, 2H), 7.15–7.10 (m, 1H), 7.07–6.97 (m, 1H), 6.56 (d, *J* = 8.8 Hz, 2H), 4.27 (brs, 2H); ^13^C NMR (150 MHz, CDCl_3_): δ 189.4, 152.7, 151.0 (dd, *J* = 252.8, 12.9 Hz), 149.1 (dd, *J* = 248.4, 12.9 Hz), 132.9 (q, *J* = 4.4 Hz), 132.5, 130.6 (q, *J* = 30.2 Hz), 129.4, 126.2, 125.5, 122.2 (q, *J* = 273.0 Hz), 118.3 (d, *J* = 18.8 Hz), 117.7 (d, *J* = 17.1 Hz), 113.9; HRMS (FAB) *m/z*: [M+H]^+^ calcd for C_16_H_11_F_5_NO 328.0761, found 328.0750.

### 3.4. (E)-3-(3,4-difluorophenyl)-1-[4-(prop-2-yn-1-ylamino)phenyl]-2-(trifluoromethyl)prop-2-en-1-one *(**9**)*


To a solution of compound **8** (67 mg, 0.2 mmol) in MeCN (0.2 mL) were added 3-bromo-1-propyne (23 µL, 0.3 mmol) and K_2_CO_3_ (85 mg, 0.6 mmol) and the mixture was stirred at 80 °C. After stirring overnight, the reaction mixture was concentrated. The residue was purified by preparative TLC eluting with toluene-EtOAc (10:1) to give compound **9** (13.4 mg, 0.037 mmol) in 18% yield as a yellow oil. Rf 0.35 (hexane/EtOAc, 3:1); UV (MeCN/H_2_O, 1:1) λ_max_ (nm) 248, 354; IR (neat) ν_max_ (cm^−1^) 3369, 3306, 3058, 2922, 1585, 1516, 1257, 1173, 1115, 915, 833; ^1^H NMR (400 MHz, CDCl_3_): δ 7.79 (d, *J* = 8.4 Hz, 2H), 7.19–7.11 (m, 2H), 7.08–6.98 (m, 2H), 6.58 (d, *J* = 8.4 Hz, 2H), 4.56 (t, *J* = 5.6 Hz, 1H), 3.98 (dd, *J* = 5.6, 2.4 Hz, 2H), 2.27 (t, *J* = 2.4 Hz, 1H); ^13^C NMR (150 MHz, CDCl_3_): δ 189.5, 152.1, 151.0 (dd, *J* = 252.8, 12.9 Hz), 150.1 (dd, *J* = 248.4, 12.9 Hz), 132.9 (q, *J* = 5.9 Hz), 132.3, 130.6 (q, *J* = 30.2 Hz), 129.4, 126.2, 125.5, 122.2 (q, *J* = 272.9 Hz), 118.3 (d, *J* = 18.8 Hz), 117.7 (d, *J* = 18.8 Hz), 112.4, 79.2, 72.2, 32.9; HRMS (FAB) *m/z*: [M+H]^+^ calcd for C_19_H_13_F_5_NO 366.0917, found 366.0913.

### 3.5. Cell Proliferation Assay Using PCa Cells

DU145 cells (5 × 10^4^) were seeded on 12-well plates (2-layer chambers) with DMEM including 5% charcoal-stripped fetal calf serum (CCS) (HyClone Laboratories, Logan, UT, USA). PC-3 cells (5 × 10^4^) were seeded on 12-well plates (2-layer chambers) with RPMI-1640 including 5% charcoal-stripped fetal calf serum (CCS) (HyClone Laboratories, Logan, UT, USA). After 24 h, cells were treated with compounds and cultured for 4 days. Medium was replaced once, at day 2 of treatment. To determine cell proliferation, cells were trypsinized and counted in triplicate using a hemocytometer. The data represent the means±SD of three replicates.

### 3.6. Antiproliferative Activity against Non-Prostate Cancer Cell Lines

Assay was performed as described previously [20]. Briefly, freshly trypsinized cell suspensions were seeded on 96-well microtiter plates at densities of 4000–11,000 cells per well with compounds. After 72 h in culture with test compounds, attached cells were fixed in 10% trichloroacetic acid and then stained with 0.04% sulforhodamine B. The absorbance at 515 nm was measured using a microplate reader (ELx800, BioTek, Winooski, VT, USA) operated by Gen5 software (BioTek, Winooski, VT, USA) after solubilizing the bound dye with 10 mM Tris base. The mean IC_50_ is the average from at least three independent experiments in duplicate.

### 3.7. Xenograft Model in Mice

Six-week-old male severe combined immunodeficient (SCID) mice were purchased from CLEA Japan (Tokyo, Japan). After an acclimatization period, 2 × 10^6^ PC-3 cells were implanted with 50% Matrigel (Corning, NY, USA) in the dorsal subcutaneous region of the SCID mice. When the tumors were large enough for their length to be measured, the mice were divided into two groups (five mice in each group) so that the mean tumor size in each group was approximately equal. For the intraperitoneal administration, chalcone **5** was administered intraperitoneally in 20 µL of DMSO at a concentration of 3 mg/kg, while the control group received 20 μL of DMSO only. For the oral administration study, chalcone **5** was administered orally in 100 μL of corn oil at a concentration of 3 mg/kg, while the control group received 100 μL of corn oil only. The intraperitoneal group was administered twice a week, and the tumor size and body weight were measured at the same time. The oral group was administered three times a week, and the tumor size and body weight were measured simultaneously using a vernier caliper and a scale, respectively. Tumor size was calculated using the following formula (length × width × width × 0.5). This animal protocol was approved by the Institutional Animal Care and Use Committee of the Graduate School of Medical Science, Kanazawa University, Kanazawa, Japan.

### 3.8. Flow Cytometric Analysis

MDA-MB-231 or KB-VIN (7 × 10^4^ cells/well) cells were seeded in a 12-well plate 24 h prior to treatment with compounds for 24 h. Compounds were used against MDA-MB-231 or KB-VIN at a concentration threefold of their IC_50_ value. Harvested and 70% EtOH-fixed cells were stained with propidium iodide (PI) containing RNase (BD Bioscience, San Jose, CA, USA) and subjected to flow cytometry (BD LSRFortessa, BD Biosciences, San Jose, CA, USA). CA-4 at 200 nM was used as a control tubulin polymerization inhibitor arresting cells at G2/M.

### 3.9. Immunostaining

MDA-MB-231 cells (2.4 × 10^4^ cells/well) were grown on an 8-well chamber slide (Lab-Tech, Waltham, USA) for 24 h prior to treatment with reagents. Cells were treated for 24 h with compound **5** or DMSO as a control for 24 h. Cells were fixed in 4% paraformaldehyde in PBS and permeabilized with 0.5% Triton X-100 in PBS. Fixed cells were labeled with mouse monoclonal antibody to α-tubulin (B5-1-2, Sigma, St. Louis, MO, USA) followed by FITC-conjugated antibody to mouse IgG (Sigma) [31]. Nuclei were labeled with DAPI (Sigma). Fluorescence labeled cells were observed using confocal microscope (LSM700, Zeiss, White Plains, NY, USA) controlled by ZEN (black edition) software (Zeiss). Confocal images were stacked and merged using ZEN (black edition) software. Final images were prepared using Adobe Photoshop.

### 3.10. Affinity Purification of Binding Proteins

Compound **9** was immobilized with Azide magnetic beads (TAS8848N1160, Tamagawa Seiki Co., Ltd. Iida, Japan) at a concentration of 125 μM according to the manufacturer’s method. PC-3 cell lysates in soluble buffer [10 mM HEPES, 150 mM Na_2_SO_4_, 1 mM EDTA, 2% CHAPS, protease inhibitor (cOmplete Mini, Roche, Basel, Switzerland)] were incubated with or without compound **5** (5 mM) at 37 °C for 1 h. The lysates (30 µL) were incubated with the compound **9**-immobilized beads (10 µL) at 37 °C for 1 h. The beads were washed five times with soluble buffer (200 µL each) and boiled at 95 °C for 5 min with SDS-PAGE sample buffer (10 µL). After the beads were removed using a magnetic stand, samples were analyzed by SDS-PAGE. The resultant gel was stained with CBB and the stained bands were cut out. The small portions of gel were treated with usual methods including reductive alkylation and in-gel digestion, and the resultant samples were analyzed with LC-MSMS (Orbitrap QE plus, Thermo Fisher, Waltham, MA, USA). The obtained peptide sequences were identified using Proteome Dicoverer software (Thermo Fisher, Waltham, MA, USA).

## 4. Conclusions

We developed seven α-trifluoromethyl chalcones and evaluated them for antiproliferative activity against androgen-independent prostate cancer cell lines. Many of them showed potent activity with submicromolar IC_50_ values. The most effective chalcone **5** also displayed significant antiproliferative activity against taxane-resistant cell lines and antitumor activity in vivo with only 3 mg/kg administration in a mouse xenograft model. To investigate the intracellular target molecules, we prepared the chemical probe **9** and confirmed that its mode of action was the same as that of chalcone **5** by using flow cytometry. Affinity purification of binding proteins from PC-3 cell lysates using compound **9**-immobilized beads revealed several candidates as the ligands for chalcone **5**. Some of them are known to stimulate cancer related cell growth.

## Data Availability

The data presented in this study are available on request from the corresponding authors.

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
