# Peer review of "α-Trifluoromethyl Chalcones as Potent Anticancer Agents for Androgen Receptor-Independent Prostate Cancer"

_molecules, 2021, doi:10.3390/molecules26092812_

Round 1

Reviewer 1 Report

The manuscript described by Saito and coworkers presents synthesis of  α-Trifluoromethyl chalcones and evaluation of their anticancer activity against DU145 and PC-3 cell lines – prostate cancer cell lines and also five additional human tumor cell lines (A549, MDA-MB-231, MCF-7, KB, KB-VIN). It is interesting work, however I have a few minor comments.

line 15 and line 117: “in vivo” sometimes is italic, sometimes not – please improve it

line 62: typo in the name 1-trifuoromethyl-1,2-benziodoxol-3(1H)- 62 one, should be 1-trifluoromethyl-1,2-benziodoxol-3(1H)- 62 one

Table 1. in first line not all words are bold – only “Cell lines/ IC50(uM) “, “Compounds” and “DU145” – please improve it

Are the synthesized compounds novel ? If yes, which ones ?

In Supplementary Information, in almost all 1H NMR spectra the integration is unreadable due to the overlap of numbers, please improve it.

Reviewer 2 Report

Well written article with interesting research on the activity of α-CF3 chalcones against androgen-independent prostate cancer lines as well as five other types of human tumor cell lines. The entire study was flawless.

Very good English.

Comments:
1) What about the novelty of chemical compounds (chalcones and compounds 1-9)? Are any of them new? Please provide relevant information.

2) Is there an explanation for the change in conformation from trans in chalcones to cis in α-CF3 chalcones? Please explain and if add relevant literature.

3) Why was it not decided to study the activity of the synthesized chalcones (not numbered in the text)?

4) Lines 72, 91/92,: the text should not be bold;

5) In all tables, please write "Compounds" (in capital letter) instead of "compounds";

6) In all tables, please add standard deviations (SD).

7) What about the Selectivity Index of the tested compounds? Are such studies planned?

8) Table 1: "Cl-DHC" - please explain right below the table, it can be with "*".

9) Lines 106/107: missing enter;

10) Line 145: The scheme description should be below it, not above.

11) Only C-4'-NH2 chalcone and its derivatives 8 and 9 were tested. What about the remaining (C-3'-NH2 and C-2'-NH2) chalcones and possible derivatives type 8 and 9? It has been shown in the literature that the activity differs depending on the position of attachment of the amino group to ring A (eg DOI: 10.3390/molecules24224129). Why, then, were only C-4'-NH2 chalcone and its derivatives selected for research?

12) In section 3: Please change "(. E)" to "(E)" everywhere. The same for the other places - change "(E)" to "(E)".

13) In section 3: Please add Rf for all compounds and melting points for solids.

14) In section 3: please add FT-IR and UV-VIS spectra (please include relevant figures in Supplementary Informations).

15) Section 3.4 - please fix - it should be normal font and not italic.

16) All chemical compound numbers should be entered in bold. Additionally, in section 3, please include their numbers next to the names of the compounds (if they are mentioned earlier in the article).

Reviewer 3 Report

Manuscript ID: molecules-1208175

          The article with title “α-Trifluoromethyl Chalcones as Potent Anticancer Agents for Androgen Receptor-Independent Prostate Cancer” by Kyoko Nakagawa-Goto et al discusses the design and synthesis of α-Trifluoromethyl chalcones. These chalcones were evaluated for their antiproliferative activities against androgen-independent prostate cancer cell lines and 5 types of human tumor cell lines. The most potent chalcone 5 (in-vitro) also showed superior antitumor activity in vivo. The mechanism of action was demonstrated  to involve cell cycle arrest (accumulation at sub-G1 and G2/M phases without interfering with microtubule polymerization). The affinity purification of binding proteins revealed several cancer cell growth-related proteins.

General comments-

          Introduction needs to provide additional references of other chalcone papers having alpha substituted olefins and their bioactivity including affinity purification or target based affinity procedures. See refs- European Journal of Medicinal Chemistry 148 (2018) 337-348; Oncotarget, 2017, Vol. 8, (No. 9), pp: 14325-14342; Journal of Medicinal Chemistry 2016 59 (11), 5264-5283 DOI: 10.1021/acs.jmedchem.6b00021; see section Group B α-substituted chalcones in Ducki etal, Anticancer Agents Med Chem. 2009 Mar;9(3):336-47. doi: 10.2174/1871520610909030336; Zhenyuan Miao et al Chem. Rev. 2017, 117, 7762−7810 DOI: 10.1021/acs.chemrev.7b00020

Although most papers above have reported their mechanism to be microtubule destabilization as mode of cell death, it was surprising to note chalcones 5 did not interfer with microtubule polymerization.

                     Section 3.2- General procedures for chalcones, the melting point for all solid chalcones should be recorded and mentioned. Is it possible to number the initial naphthalene and benzothiophene derived chalcones in the write up?

          In the NMR  data for (E)-3-(3,4-difluorophenyl)-1-(naphthalen-1-yl)prop-2-en-1-one the second doublet for olefinic double bond (J = 16 Hz) couldn’t be found/ is missing. The supplementary information file spectra was not helpful to locate the peaks.

How was the purity of chalcones checked before biological evaluation?

The reason for synthetic use of amino chalcones was not mentioned for further introduction of propargyl group.

                   Please check the references section for correctness and include the  DOI for all references.

Round 2

Reviewer 2 Report

Thank you for all the answers and explanations.

My final comments:
1) Section 3.1. Please add on what equipment the melting points were measured and the FT-IR and UV-VIS spectra were made.
2) In Supplementary Informations: in the titles of FT-IR spectra, please add the technique in which the spectra  were made ("neat") and in the titles of the UV-VIS spectra, please add what solvent was used (MeCN/H2O, 1:1). This will make it easier to read the SI without searching in the text of the article.

After following these recommendations, the article will meet all the requirements to be published in Molecules MDPI.
